# Modelling of Dynamic Behaviour in Magnetic Nanoparticles

**DOI:** 10.3390/nano11123396

**Published:** 2021-12-15

**Authors:** Max Tigo Rietberg, Sebastiaan Waanders, Melissa Mathilde Horstman-van de Loosdrecht, Rogier R. Wildeboer, Bennie ten Haken, Lejla Alic

**Affiliations:** Magnetic Detection & Imaging Group, Technical Medical Centre, University of Twente, 7522 NH Enschede, The Netherlands; m.t.rietberg@student.utwente.nl (M.T.R.); s.waanders@utwente.nl (S.W.); m.m.vandeloosdrecht@utwente.nl (M.M.H.-v.d.L.); r.r.wildeboer@alumnus.utwente.nl (R.R.W.); b.tenhaken@utwente.nl (B.t.H.)

**Keywords:** modelling, magnetic nanoparticles, Néel relaxation, Brownian relaxation, Fokker-Planck equation, particle response function, anisotropy

## Abstract

The efficient development and utilisation of magnetic nanoparticles (MNPs) for applications in enhanced biosensing relies on the use of magnetisation dynamics, which are primarily governed by the time-dependent motion of the magnetisation due to externally applied magnetic fields. An accurate description of the physics involved is complex and not yet fully understood, especially in the frequency range where Néel and Brownian relaxation processes compete. However, even though it is well known that non-zero, non-static local fields significantly influence these magnetisation dynamics, the modelling of magnetic dynamics for MNPs often uses zero-field dynamics or a static Langevin approach. In this paper, we developed an approximation to model and evaluate its performance for MNPs exposed to a magnetic field with varying amplitude and frequency. This model was initially developed to predict superparamagnetic nanoparticle behaviour in differential magnetometry applications but it can also be applied to similar techniques such as magnetic particle imaging and frequency mixing. Our model was based upon the Fokker–Planck equations for the two relaxation mechanisms. The equations were solved through numerical approximation and they were then combined, while taking into account the particle size distribution and the respective anisotropy distribution. Our model was evaluated for Synomag^®^-D70, Synomag^®^-D50 and SHP-15, which resulted in an overall good agreement between measurement and simulation.

## 1. Introduction

Magnetic nanoparticles (MNPs) have become a popular research subject in biomedicine thanks to their high biocompatibility, long shelf life and straightforward logistics when compared to radioactive agents for similar applications. The biomedical application of MNPs ranges from therapy, such as in magnetic hyperthermia or targeted drug delivery [1,2], to diagnostics, where they are applied as contrast agents or tracers [3,4], or even theranostics [5]. Sensing techniques that employ MNPs include AC magnetometry [6], differential magnetometry [7,8], magnetic particle spectroscopy (MPS) [9], and magnetic particle imaging (MPI) [10]. All of these techniques rely on the targeted magnetic manipulation and accurate acquisition of the dynamic response of an individual MNP. Therefore, an accurate model of the dynamics governing their magnetic properties to enhance the sensing techniques is of vital importance. However, sensing technologies are often developed sub-optimally regarding magnetisation dynamics. The options available for performance optimisation include (for example) an improved signal-to-noise ratio (SNR) and optimised excitation sequences. The main goals of this model are:To model the behaviour of particles: allow for the optimisation of particles for a given application without the need for extensive empirical testing;To predict a particle’s properties, magnetic field properties and environmental parameters, such as viscosity, based on the behaviour of the MNPs.

In recent years, many models have been developed to describe the individual aspects of MNP magnetisation dynamics under certain conditions, including heat dissipation [11], harmonic field response [12,13,14,15], viscosity effects [16], temperature dependence [17], core distribution [18], one dimension [19] and damping of the magnetic field [20]. After Brown’s seminal paper [21], the characteristic magnetic relaxation times were assessed for a particular case with a constant magnetic field under a step function regime [22]. However, the dynamic behaviour of MNPs in changing magnetic fields is complex, especially in the domain where simultaneous Brownian and Néel processes take place. Brownian relaxation aligns the whole particle with the magnetic field, while Néel relaxation aligns the internal magnetic dipole within the particle. The most frequently used approach to model MNP behaviour under conditions of varying magnetic fields currently involves a phenomenological model of the magnetisation response using the steady-state approximation of magnetic particles rotating toward the field’s orientation [7,23,24,25]. However, a significant downside to this approach is the fact that particle anisotropy and time-delay effects are ignored; therefore, this approach often does not hold in practice.

The known relaxation mechanisms (Brownian and Néel) have been modelled using two separate Fokker–Planck equations (FPEs) [22]. The magnetisation dynamics of the spherical particles with a non-critical diameter (meaning that either Brownian or Néel relaxation is dominant) can be well described with these equations, but not of particles with a critical diameter (no dominant mechanism) because the FPEs are separate and lack connection. Consequently, this publication presents a practical and effective way of simultaneously solving these FPEs, which accurately describes the non-linear magnetisation dynamics of various superparamagnetic nanoparticles surprisingly well for non-spherical particles. Its outcomes were validated with magnetometer measurements of three different types of MNP. This model has potential as a tool for use in the design and validation of optimised MNPs for biomedical applications. Furthermore, this enables the tailored adjustments of new sensing devices to match the MNP characteristics and consequently to maximise sensitivity.

## 2. Theory

The behaviour of monodisperse (meaning that the mixture only contains MNPs of the same size) superparamagnetic MNPs of non-critical diameter can be described by solving a set of equations that capture Brownian and Néel relaxation. The probability distribution of MNP magnetic moments is defined as F(x,t), where θ is the polar angle between the MNP dipole moment and the direction of the driving magnetic field, x≡cosθ and *t* is the time. This set of equations (which is approximated by Equations (Equation 1) and (Equation 2)) is referred to as the Fokker–Planck equations (FPEs). These FPEs are first-order non-linear partial differential equations (PDE) that capture the time evolution behaviour of a probability density function describing transient convection–diffusion with a quadratic space-dependent diffusion and time-dependent driving magnetic field [22]: (1)$$\begin{array}{cc} \frac{\partial }{\partial t}F= & \frac{1}{2\tau_{B}}\frac{\partial }{\partial x}\left[\left(1-x^{2}\right)\left(\frac{\partial }{\partial x}F-\xi \left(t\right)F\right)\right] \end{array}$$(2)$$\begin{array}{cc} \frac{\partial }{\partial t}F= & \frac{1}{2\tau_{N}}\frac{\partial }{\partial x}\left[\left(1-x^{2}\right)\left(\frac{\partial }{\partial x}F-\xi \left(t\right)F-2\sigma xF\right)\right] \end{array}$$

The driving field B(t) (with varying amplitude and frequency) was described using the effective field parameter $\xi =\left(m_{0}/k_{B}T\right)B\left(t\right)$ and the particle anisotropy constant *K* was described by the parameter $\sigma =KV_{c}/k_{B}T$. Each particle was characterised by a constant magnetic dipole moment with a magnitude of $m_{0}=M_{s}V_{c}$, with $M_{s}$ for the saturation magnetisation and $V_{c}$ for the volume of a magnetic core. $k_{B}$ is Boltzmann’s constant, and *T* is temperature. Furthermore, the relaxation times $\tau_{B}$ and $\tau_{N}$ represent the effective characteristic time constant for Brownian and Néel relaxation, respectively, and read:(3)$$\begin{array}{cc} \tau_{B} & \equiv \frac{3\eta V_{h}}{k_{B}T} \end{array}$$(4)$$\begin{array}{cc} \tau_{N} & \equiv \frac{V_{c}\left(1+\alpha^{\prime 2}\right)M_{s}}{2\gamma_{e}\alpha^{\prime }k_{B}T} \end{array}$$

Here, η is medium viscosity, $\alpha^{\prime }$ is the damping constant, $\gamma_{e}$ is the electron gyromagnetic ratio and $V_{h}$ is the hydrodynamic volume of a particle submerged in medium.

Without a known analytical solution, under adiabatic approximation and solving for space-dependent diffusion, the FPEs reduce to the well-known Langevin function [26,27]. Although this is an elegant solution, the Langevin function does not offer an accurate description of superparamagnetism, especially at a frequency range where both relaxation processes are equally important. Consequently, the Langevin function fails to accommodate the influence of anisotropy and particle–particle interactions.

Another numerical pathway for solving F(x,t) is by approximating the space-dependent diffusion using Legendre polynomials [13]:(5)$$F\left(x,t\right)=\sum_{l=0}^{\infty }a_{l}\left(t\right)P_{l}\left(x\right)$$

Substituting this approximation into Equations (Equation 1) and (Equation 2) results in a new set of ordinary differential equations (ODEs) for Brownian and Néel relaxation, respectively [22]: (6)$$\begin{array}{cc} \frac{2\tau_{B}}{l(l+1)}\frac{da_{l}}{dt}= & -a_{l}+\xi \left(t\right)\left[\frac{a_{l-1}}{2l-1}-\frac{a_{l+1}}{2l+3}\right] \end{array}$$(7)$$\begin{array}{c} \begin{array}{cc} \frac{2\tau_{N}}{l(l+1)}\frac{da_{l}}{dt}= & -a_{l}+\xi \left(t\right)\left[\frac{a_{l-1}}{2l-1}-\frac{a_{l+1}}{2l+3}\right]\\ \; & +\sigma \left[\frac{\left(l-1\right)a_{l-2}}{\left(2l-3)(2l-1\right)}+\frac{la_{l}}{\left(2l-1)(2l+1\right)}\right.\\ \; & \left.-\frac{\left(l+1\right)a_{l}}{\left(2l+1)(2l+3\right)}--\frac{\left(l+2\right)a_{l+2}}{\left(2l+3)(2l+5\right)}\right] \end{array} \end{array}$$

These ODEs can be used to calculate the time average of *x* (again, x≡cosθ) which is correlated with the magnetic moment:(8)$$\begin{array}{cc} \langle x(t)\rangle = & \frac{2}{3}a_{1}\left(t\right) \end{array}$$(9)$$\begin{array}{cc} \frac{d}{dt}M\left(t\right)= & nM_{s}V_{c}\frac{d}{dt}\langle x\left(t\right)\rangle \end{array}$$

However, this approach does not combine Brownian and Néel relaxation, and results in two separate magnetisation curves. The common practice to omit this problem for static fields and relaxation processes is to only consider the dominant relaxation mechanism [28,29,30], or in the critical size range (where both processes are equally contributing), use the geometric mean of both relaxation times [11]. However, neither of these practices reflect reality. Alternative attempts have been made to describe the particle response in terms of a superposition of both relaxation processes [29,31]. When the applied field rapidly changes (e.g., in the case of AC magnetometry), a simple superposition fails to describe the magnetic behaviour of the particles. This can be partially attributed to the inaccurate assumption that these processes are fully independent.

## 3. Methods

### 3.1. MNP Samples

Three different types of superparamagnetic MNPs were used to acquire the particle response functions that are necessary to validate the model: Synomag^®^-D70, Synomag^®^-D50 (micromod Partikeltechnologie GmbH, Rostock, Germany) and SHP-15 (Ocean Nanotech, San Diego, CA, USA). The first two are nanoflower-shaped particles, while the latter is a ‘normal’ particle; as can be seen in Figure 1. It has to be noted that the model was developed with spherical particles in mind, which does not reflect the real-world properties of the Synomag particles. Table 1 gives an overview of the characteristics for all three MNPs, which are polydisperse (meaning that the mixture contains MNPs of varying size, instead of only MNPs with the same size) with an anisotropy constant dependent on size (see Equation (Equation 13)). The core diameter (spread) was determined using an analysis of TEM micrographs, while the hydrodynamic diameter was determined using number-based dynamic light scattering. All of our samples consisted of 140 µg iron dissolved in water, resulting in a total volume of 140 µL contained in glass vials which were kept at room temperature.

### 3.2. Data Acquisition for Experimental Observations

The particle response function (PRF) was acquired using the characterisation mode of the superparamagnetic quantifier (SPaQ), which is an in-house developed magnetometer utilising a homogeneous magnetic field [35]. PRFs were assessed by exposing the samples to a continuous alternating magnetic field ($B_{ac}=$ 1.5 mT, frequency = 2.5 kHz), with an increasing offset field $B_{+}$ ranging from −24.2 to 24.2 mT:(10)$$B\left(t\right)=B_{AC}\sin\left(2\pi ft\right)+{\dot{B}}_{+}t$$

The subsequent magnetisation signal is acquired by a set of gradiometric coils with a sensitivity of $S_{det}=$ 37.8 mT/A, which leads to an induced voltage $U_{det}\left(t\right)/S_{det}=-\frac{d}{dt}M\left(t\right)$.

### 3.3. Model

We approach the fact that we are dealing with two separate relaxation mechanisms by initially considering both processes to operate independently. Since a magnetometer measures the *rate of change* of the magnetisation, one observes the sum of two orthogonal rotations. Following the Legendre approximations (as described in Equations (Equation 6) and (Equation 7)) [22,36], the contribution of Néel and Brownian relaxation processes to the response of the MNPs to an externally applied magnetic field is assessed by
(11)$$\frac{d}{dt}M\left(t\right)=\sqrt{{\left(\frac{d}{dt}M_{Brown}\right)}^{2}+{\left(\frac{d}{dt}M_{N\overset{´}{e}el}\right)}^{2}}$$

The initial theoretical assumption of monodispersity in MNPs does not match the current reality of commercially available polydisperse MNPs. Therefore, to accurately model particle response function (solution of Equation (Equation 11)), the particle size distribution needs to be taken into account. Consequently, we approximated the polydispersity in core diameters $d_{c}$ by a normal distribution, which is detailed in Table 1. For numerical purposes, this distribution is discretised into an increasing number of bins until the resulting individual MNPs response (again, solution of Equation (Equation 11)) stabilises, which means that a further increase in bin density does not noticeably change the solution. Finally, the PRF is defined as the weighted average of these responses according to the discretised normal distribution.

Brownian relaxation influences the Néel relaxation by orienting the MNPs along the direction of the applied magnetic field [24,37]. If Brownian relaxation is not possible (e.g., when particles are trapped in a medium or tissue), then Brownian relaxation is prohibited. Depending on the orientation of the magnetic easy axes of the particles suspended in the sample under investigation, this effect alters the Néel relaxation behaviour of the particles if their anisotropy is not equal to unity (i.e., they deviate from perfect spherical symmetry). Following initial research by Shliomis et al. [38], this effect is modelled by an effective anisotropy constant as an energy term $K_{\mathrm{eff}}$, which is composed of both longitudinal and transverse anisotropy energies. Assuming the potential landscape as $U=K\sin^{2}\theta$, then we have $U_{\|}=K$ in the longitudinal case and $U_{⊥}=0$ for both transverse orientations. This results in an effective anisotropy constant:(12)$$K_{\mathrm{eff}}=\frac{1}{3}K_{\|}+\frac{2}{3}K_{⊥}=\frac{1}{3}K$$

Considering the fact that the anisotropy constant changes with the particle core size [39], a polydisperse sample cannot be modelled using only one anisotropy constant. Therefore, a relation is proposed, which results in a different anisotropy constant for each core size:(13)$$K=K_{a}d_{c}+K_{b}$$
where $K_{a}$ and $K_{b}$ are fit parameters for known anisotropy constants for certain diameters. The resulting *K* can be filled into Equation (Equation 12) to obtain the effective constant. While Equation (Equation 13) is not a perfect solution due to the assumption of linear relation (which would determine the anisotropy constant for every particle size), it is an improvement from the same constant for all particle sizes.

To evaluate the formulated magnetisation dynamics (Equation (Equation 11)) and to elaborate on the regimes in which either relaxation mechanism might dominate, we computed the resultant magnetisation curves for monodisperse nanoparticle samples of selected core sizes ($d_{c}$ = 10 nm, 18 nm, 26 nm). This was visualised by means of their PRFs, which are similar to the derivative of the magnetisation curve or the point spread function in MPI. This denotes the sample’s signal amplitude as a function of the applied field magnitude.

### 3.4. Model Validation

Our model was validated by comparing the results to a simplified solution of FPE (namely the Langevin equation) and experimental observations. Three types of MNPs, namely Synomag^®^-D70, Synomag^®^-D50 and SHP-15, were evaluated for their magnetic performances using their PRF. This was repeated three times and then averaged to ensure reproducibility. These results were compared with a numerical evaluation of the model that we introduced earlier. It is common practice to set the damping constant to 0.1, and work with a ferrofluid viscosity of η=1.0049 mPa s; the other parameters are defined in Table 1. The model was evaluated in MATLAB (2021a, MathWorks, Natick, MA, USA) using the *ode15s* subroutine, a variable-step, variable-order solver for stiff differential equations based on the numerical differentiation formulas. The Legendre expansion converges fairly rapidly, and the set of ODEs was evaluated up to the 60th coefficient.

To quantify the goodness of fit, the full-width at half-maximum (FWHM) and the mean of absolute residuals (MoR) were used. The FWHM is an important characteristic in MPI because it denotes the spatial resolution. The MoR is the mean of the absolute difference inside the FWHM window. Thus, the difference between the experimental data and the model result was calculated inside the FWHM window. The absolute value of these differences was averaged to obtain the MoR: $(\sum_{n}|M\left(B_{n}\right)-E\left(B_{n}\right)|)/n$, where *M* is the model result and *E* is the experimental data. Due to current technical limitations, the offset field was limited to 24.2 mT which was insufficient to reach the FWHM of the SHP-15. Therefore, for SHP-15, our model was validated using a slightly extrapolated data by fitting a normal distribution on the experimental data.

## 4. Results

### 4.1. Numerical Modelling of Brownian and Néel Dominated M–H Curves

To explore the boundaries of the developed model, we calculated the behaviour of particles with characteristics that would, under normal circumstances, lead to either Brownian or Néel-dominated magnetisation behaviour. Figure 2 illustrates the magnetisation curves for iron oxide particles of different core diameters (10 nm, 18 nm and 26 nm) and a constant coating thickness (6 nm). As expected, the largest particle shows a magnetisation curve corresponding to Brownian-dominated relaxation, while the smallest particle shows a magnetisation curve corresponding to Néel behaviour. However, the 18 nm particle does not have a dominant relaxation mechanism and shows the competing behaviours of both relaxation mechanisms. We observed the Néel behaviour for the low-offset field ($B^{+}$) values, whereas Brownian relaxation dominates for higher fields. This relates well to observations by Deissler et al. [22], who also showed a transition from Néel to Brownian behaviour for increasing field strengths.

### 4.2. Experimental Verification of Particle Response Functions

The experimental and numerical results for SHP-15, Synomag^®^-D50 and Synomag^®^-D70 are shown in Figure 2. The data are normalised with respect to the largest value to assess shape similarity. Quantification of the goodness of fit can be seen in Table 2. Overall, we observed good agreement between the shape predicted by the simulations and the experimental results, showing almost exclusively Brownian relaxation in the case of Synomag^®^-D70 and Néel relaxation for SHP-15. Synomag^®^-D50 shows a combination of Brownian and Néel relaxation. Furthermore, for SHP-15, the Langevin function fails to adequately predict the shape of the PRF.

## 5. Conclusions and Discussion

In this work, we explored the magnetisation dynamics of a variety of MNPs. The FPEs pertaining to Brownian and Néel relaxation were solved by means of their Legendre approximations. A strong point of our model is the fact that it includes the effects of polydispersity and the resulting anisotropy. Our model also simultaneously demonstrates the impact of both relaxation mechanisms on the magnetisation dynamics of particles. We observed a deviation from the commonly used Langevin solution for the magnetic behaviour of MNPs, even in the case of larger particles that predominantly relax through Brownian relaxation. For all cases, it was observed that the adiabatic approximation (the Langevin equation) was not valid because of the finite Brownian relaxation time, even at the relatively low frequency of 2.5 kHz and low excitation field strengths. Moreover, we found that the PRF shape of MNPs in the critical size range was well predicted by a combined model that takes both Brownian an Néel relaxation into account. The difference between the model result and the experimental data might be explained by the assumption of a spherical particle, as the Synomag particles are instead flower-shaped. Furthermore, Brownian relaxation dominates in the high field range, while Néel relaxation describes the low-field regime quite well. Keeping this effect in mind, a close look at the PRFs obtained by Arami et al. [40] leads to a similar conclusion. Here, an increase in the steepness of PRFs was observed for increasing viscosity, while particles suspended in chloroform (lowering the viscosity) showed a much flatter PRF, all else being equal.

The results presented in this work are qualitatively well described by a system of independent Brownian and Néel relaxation, despite the inevitable simplification of details such as Brownian alignment, which influences the Néel process. For example, the Brownian relaxation process influences the Néel relaxation through the alignment of the particle’s magnetic easy axis, which is hindered when particles are immobilised. This effect can be easily corrected for in this extreme case by defining an effective anisotropy constant for immobilised particles with randomly oriented easy axes. It would be most interesting to measure the PRFs for particles immobilised under application of a strong external magnetic field and for particles immobilised in zero field, and then verify this hypothesis.

Following the analogy used by Weizenecker et al. [41], our model could likely be improved by modelling with the coupled FPE instead of the currently used decoupled FPEs. This coupled model was based on a coupled Fokker–Planck equation but has not yet been validated through experimental data. Moreover, as noted in previous sections, the linear relation between anisotropy and (only) the particle size is limited. This imperfect solution is a good start, however, because particle size is one of the factors that affect the anisotropy constant the most.

We must also comment that the experimental variation in particle parameters $d_{c}$, $d_{h}$, *K* and $M_{s}$, as well as the inherent uncertainty in iron concentration complicates truly quantitative matching between the model and experiment. This deserves more attention, especially in view of the considerable variety in the reported anisotropy constants *K* in the literature [32,42,43,44]. The current values for $K_{a}$ and $K_{b}$ are based on *K* of Ludwig et al. because this value lies in the middle of the range found in the literature and fits our results the best. Nevertheless, by means of designing new or improving existing particles, it is possible to fine-tune the desired PRF parameters (e.g., FWHM) because the connection between particle composition and magnetisation dynamics can be better understood by studying this model. This will in turn improve the quality of biomedical applications.

## Figures and Tables

**Figure 1 nanomaterials-11-03396-f001:**
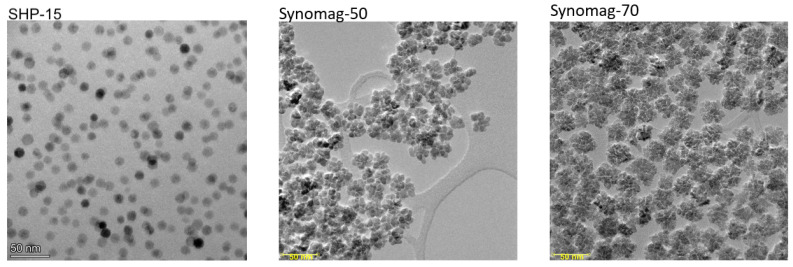
TEM micrographs acquired at an accelerating voltage of 300 kV: SHP-15 (**left**) was analysed using FEI Titan Cubed, Synomag-50 (**middle**) and Synomag-70 (**right**) were analysed using Philips CM300ST.

**Figure 2 nanomaterials-11-03396-f002:**
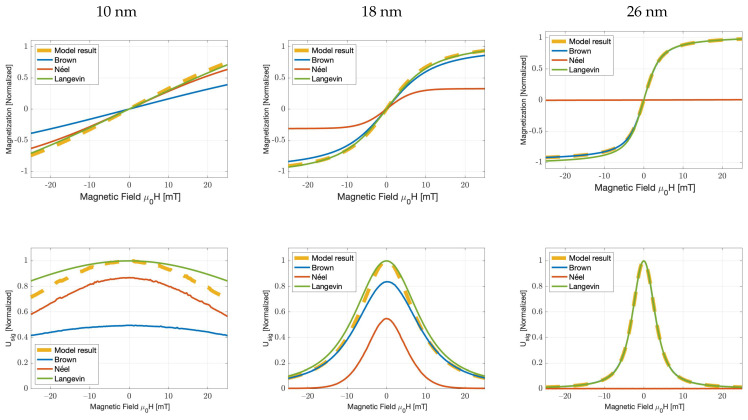
Magnetisation curves, their numerical derivatives and corresponding relaxation times as a function of the magnetic field for monodisperse particles, obtained from the numerical evaluation of the Brownian and Néel FPEs for 10 nm (**left**), 18 nm (**middle**) and 26 nm (**right**) particles. Simulation parameters: K=20 kJ/m${}^{3}$, $d_{h}=d_{c}+12$ nm, T=300 K, $\alpha^{\prime }=0.1$, η=1.0049 mPas, $M_{s}=300$ kJ/m${}^{3}$T.

**Figure 3 nanomaterials-11-03396-f003:**
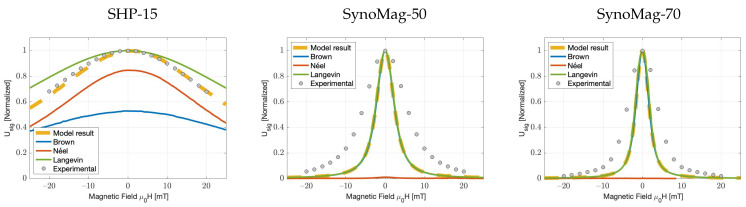
Particle response functions (experimental and simulated) for three particle-types under the application of a 2.5 kHz, 1.3 mT/$\mu_{0}$ AC field, $\alpha^{\prime }=0.1$, η=1.0049 mPas, the particles’ characteristics can be found in Table 1. (**Left**): SHP-15, T: 298 K, bins: {10, 11,..., 14} nm. (**Middle**): Synomag^®^-D50, T: 267K, bins: {20, 21, ..., 28} nm. (**Right**): Synomag^®^-D70, T: 267K, bins: {22, 23,..., 36} nm.

**Table 1 nanomaterials-11-03396-t001:** MNP parameters: $d_{c}$: core diameter; $d_{h}$: hydrodynamic diameter; $K_{a}$ and $K_{b}$: anisotropy constants based upon Equation (Equation 13) [32]; and $M_{s}$: saturation magnetisation [33,34].

	$d_{c}$	$d_{h}$	Shell	$K_{a}$	$K_{b}$	$M_{s}$
	(nm)	(nm)	(kJ m${}^{-3}$ nm${}^{-1}$)	(kJ m${}^{-3}$)	(kA m${}^{-1}$)	
SHP-15	12.20±1.23	35.00±10.04	$d_{c}+22.80$	0.150	5.0	205
Synomag-D50	24.30±3.17	29.87±8.23	$d_{c}+5.57$	0.150	9.5	420
Synomag-D70	29.00±4.00	45.90±13.64	$d_{c}+16.90$	0.150	9.5	420

**Table 2 nanomaterials-11-03396-t002:** Quantification of the goodness of fit of Figure 3, based on difference in full-width at half-maximum from the experimental data and the mean of absolute residuals in the FWHM window (the closer it is to 0, the better it is).

	SHP-15 (Extrapolated)		Synomag^®^-D50		Synomag^®^-D70	
	FWHM (% diff)	MoR	FWHM (% diff)	MoR	FWHM (% diff)	MoR
Model	11.3	0.02	−56.7	0.30	−55.1	0.30
Langevin	49.5	0.07	−55.2	0.29	−51.2	0.27

## Data Availability

The MATLAB-code and experimental data relating to this research were made available: 10.4121/14900565.

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
