# Peer review of "Modelling of Dynamic Behaviour in Magnetic Nanoparticles"

_nanomaterials, 2021, doi:10.3390/nano11123396_

Round 1

Reviewer 1 Report

All my previous concerns have been addressed by the authors.The paper has improved its overall quality and can be accepted for publication.

Reviewer 2 Report

The authors made all the corrections as raised. Thus, it can be accepted in its present form. 

This manuscript is a resubmission of an earlier submission. The following is a list of the peer review reports and author responses from that submission.

Round 1

Reviewer 1 Report

In this paper, the author develop an approximation to model and evaluate its performance for MNPs exposed to a magnetic field with varying amplitude and frequency . I feel that the authors need to first address quite a few concerns of mine, before this manuscript can be considered for publication. 

1. Please introduce the optimization effect of your model in the applications mentioned above?

2. Does your model proposed in the article have a relevant theoretical basis or derivation process?

3. As far as I know, Brownian relaxation only exists in liquid. Does your model apply to magnetic particles in other material states?

4. “model and evaluate its performance for MNPs exposed to a magnetic field with varying amplitude and frequency” is mentioned in abstract, but the experimental part does not reflect the evaluation effect of your model on the variable frequency.

Reviewer 2 Report

The manuscript entitled, ‘Modelling magnetic nanoparticles using combined Néel and
Brownian relaxation
’ reports a detailed modeling of magnetic nanoparticles with dominance of Néel and Brownian relaxation. The topic is quite interesting but there is some lack of some information. I am mentioning some to correct before submission.

  1. Néel relaxation is affected for restricted/arrested nanoparticles in polymer matrices. How the author did consider it?
  2. Did the author compromise the blocking behavior of the nanoparticles? How the blocking temperature affects the relaxation procedure?
  3. I am recommending some articles for elaborating the references and recent works. Those should be included for betterment of the literature survey: https://doi.org/10.1016/j.biotechadv.2020.107611;https://doi.org/10.1063/1.1801687;https://doi.org/10.1002/pat.5344; https://doi.org/10.1016/j.jmmm.2020.167538.
  4. How did the author consider the size based magnetic relaxation for magnetic nanoparticles?   
  5. Is the crystal structure affects this behavior and modeling a lot? Please argue.
  6. The introduction part is not clearly written. That should be improved. There is several lack of clear justification of sentences.
  7. The aim/goal is not clear written in the introduction last paragraph why this kind of modeling is required? And also the author should write about the versatility of this kind of modeling.
  8. What does the author silicified in the text as ‘biological imaging’? What type of imaging is written here? Because there are several restriction of property-morphology relationship that could affect the relaxation process especially for different type of MRI or PET techniques.
  9. What about the agglomeration effect of nanoparticles in relaxation behavior?

Reviewer 3 Report

The manuscript is devoted to the development of a new model to predict, design and validate the performance of the magnetic nanoparticles exposed to a alternating magnetic field. This model is based on Fokkler Planck equations, for the Neel and Brown mechanisms, and the effect of the particle size distribution and the respective anisotropy distribution is taken into account. The model is evaluated in Synomag®-D70, Synomag®- D50 and Perimag® samples.

The manuscript is clear and well organized. However, in my opinion, the manuscript doesn’t provide a compressive analyze of the topic. Additionally, the novelty of this work is very limited; it is not very clear what is new knowledge can be provided by this study in field of magnetic nanoparticles performance design and optimization. In the next paragraphs my comments and suggestions regarding this paper can be found:

  1. One of the big issues of the work is the lack of the characterization of the samples employed in the work. The experimental parameters of the samples used for the simulations (shown in Table 1) are taken for other works. Although authors can think that this is valid because they are commercial particles, in my personal opinion, having a characterization of the samples is vital to make the work accurate. Additionally, the sample’s parameters are taken from different works, so characterization parameters of different particles are being mixed.
  2. Some of the data (Table 1) obtained from the referenced works is not shown in that works. For example, in reference [29] the core diameter (dc) of Synomag-D50 is 24 whereas in the present papers is 24.3 nm.
  3. The name of the samples employed in reference [28] and the name of the samples of the present paper are different. For example, in [28] the samples are Nanomag-MIP and Resovist and in the present paper are Perimag and Synomag. So, the reader should assume that Perimag and Nanomag-MIP samples are the same ones, based on the size of the sample. This makes difficult to follow the work and also to verify where the data of Table 1 comes from.

In order to avoid all this issues and to improve the results I would recommend the author to do a deep characterization of the samples employed for the work using different techniques as TEM and VSM, among other. This would improve a lot the quality of the work. This characterization section can be included in the supporting information if the author don’t want to include it in the main paper.

4. As it is mentioned by the author (lines 245-246) a huge variety of anisotropy constant are being reported in literature. So, what criteria has the author follow to choose the K of Ludwig et al. paper [30] in Table 1? This is not mentioned in the paper and it is very important point.

5. One of the most tricky point in the paper is the anisotropy issue. The author proposed the equation (13) for including the anisotropy constant of a polydisperse sample. However, this samples are nanoflowers and cluster. How is the arrangement effect included in the anisotropy constant? Would it be different if we would have non-interacting particles of same size of the cluster or nanoflowers? Would the Brown relaxation be the same in both type of particles (non-interacting vs interacting)?

In my opinion, a deeper analysis of the anisotropy needs to be carried out and the effect of a contribution arises from dipolar interactions need to be mentioned and modeled. Only to consider the particle size in the anisotropy in this kind of particles is limited.

  1. Regarding the results and as it is mentioned by the author, in the second graphic of the Figure 3 the model proposed by the authors doesn’t show a good agreement with the experimental results. In my opinion this is strong evidence that the model needs to be improved before being published. Where does the mismatch come from? This part is barely mentioned in the paper.

For these reasons, I cannot recommend the publication of this manuscript in the Nanomaterials journal.